# Potential of Smoke-Water and One of Its Active Compounds (karrikinolide, KAR_1_) on the Phytochemical and Antioxidant Activity of *Eucomis autumnalis*

**DOI:** 10.3390/antiox8120611

**Published:** 2019-12-03

**Authors:** Adeyemi Oladapo Aremu, Nqobile Andile Masondo, Jiri Gruz, Karel Doležal, Johannes Van Staden

**Affiliations:** 1Research Centre for Plant Growth and Development, School of Life Sciences, University of KwaZulu-Natal Pietermaritzburg, Private Bag X01, Scottsville 3209, South Africa; masondo@sun.ac.za; 2Indigenous Knowledge Systems (IKS) Centre, Faculty of Natural and Agricultural Sciences, North West University, Private Bag X2046, Mmabatho 2735, South Africa; 3Department of Botany and Zoology, Stellenbosch University, Stellenbosch, Private Bag X1, Matieland 7602, South Africa; 4Laboratory of Growth Regulators, Faculty of Science, Palacký University & Institute of Experimental Botany AS CR, Šlechtitelů 11, CZ-783 71 Olomouc, Czech Republic; jiri.gruz@gmail.com (J.G.); karel.dolezal@upol.cz (K.D.); 5Department of Chemical Biology and Genetics, Centre of the Region Haná for Biotechnological and Agricultural Research, Faculty of Science, Palacký University, Šlechtitelů 27, CZ-783 71 Olomouc, Czech Republic

**Keywords:** asparagaceae, conservation, eucomic acid, flavonoids, hydroxybenzoic acids, hydroxycinnamic acids, phenolic acids, micropropagation

## Abstract

*Eucomis autumnalis* (Mill.) Chitt. subspecies *autumnalis* is a popular African plant that is susceptible to population decline because the bulbs are widely utilized for diverse medicinal purposes. As a result, approaches to ensure the sustainability of the plants are essential. In the current study, the influence of smoke-water (SW) and karrikinolide (KAR_1_ isolated from SW extract) on the phytochemicals and antioxidant activity of in vitro and greenhouse-acclimatized *Eucomis autumnalis* subspecies *autumnalis* were evaluated. Leaf explants were cultured on Murashige and Skoog (MS) media supplemented with SW (1:500, 1:1000 and 1:1500 *v*/*v* dilutions) or KAR_1_ (10^−7^, 10^−8^ and 10^−9^ M) and grown for ten weeks. In vitro regenerants were subsequently acclimatized in the greenhouse for four months. Bioactive phytochemicals in different treatments were analyzed using ultra-high performance liquid chromatography (UHPLC-MS/MS), while antioxidant potential was evaluated using two chemical tests namely: DPPH and the *β*-carotene model. Smoke-water and KAR_1_ generally influenced the quantity and types of phytochemicals in in vitro regenerants and acclimatized plants. In addition to eucomic acid, 15 phenolic acids and flavonoids were quantified; however, some were specific to either the in vitro regenerants or greenhouse-acclimatized plants. The majority of the phenolic acids and flavonoids were generally higher in in vitro regenerants than in acclimatized plants. Evidence from the chemical tests indicated an increase in antioxidant activity of SW and KAR_1_-treated regenerants and acclimatized plants. Overall, these findings unravel the value of SW and KAR_1_ as potential elicitors for bioactive phytochemicals with therapeutic activity in plants facilitated via in vitro culture systems. In addition, it affords an efficient means to ensure the sustainability of the investigated plant. Nevertheless, further studies focusing on the use of other types of antioxidant test systems (including in vivo model) and the carry-over effect of the application of SW and KAR_1_ for a longer duration will be pertinent. In addition, the safety of the resultant plant extracts and their pharmacological efficacy in clinical relevance systems is required.

## 1. Introduction

For centuries, indigenous people have recognized the role of fire and smoke on plant cultivation and productivity [1,2]. In an attempt to explore this knowledge under standard conditions, scientists have simulated this phenomenon by generating smoke in a drum using compressed air and bubbling through distilled water to form smoke-water (SW). Smoke and fire (smoke-technology) holds great potential in various agricultural and biological fields, and the scientific evidence on their positive role has been demonstrated in several plants [3]. The active compound was successfully isolated and identified as karrikinolide (KAR_1_), previously referred to as butenolide; this has resulted in its increased interest by researchers globally [1,2,3,4,5,6]. Given that no two batches of SW contain exactly the same balance or concentration of compounds, the isolation of active compounds in SW eliminates the disparity and ambiguity often associated with SW in crude solution [7]. Presently, several types of KAR, generally referred to as karrikins, have been identified, and are recognized as a new family of plant growth regulators (PGRs) [8]. Both SW and KAR_1_ are known to interact with other PGRs [8] and often exhibit cytokinin and auxin-like activities, as demonstrated in the mungbean bioassay [9]. The use of different PGRs, especially cytokinins, has been demonstrated to be a vital elicitor of valuable phytochemicals in medicinal plants [10,11,12]. Thus, SW and karrikins hold great potential as useful tools for enhancing plant productivity, given their influence on plant growth and development, as well as on biochemical pathways, including the phenylpropanoid pathway that serves as a rich source of metabolites in plants [2,13,14,15]. In recent times, phytochemicals have received increasing attention due to their antioxidant properties and ability to counteract oxidative stress associated with various diseases [16]. As a result, these bioactive phytochemicals have been widely-explored for their therapeutic and pharmacological (e.g., antioxidant, antimicrobial, anti-inflammatory, and anti-diabetics) value, as evidenced in several African medicinal plants [17].

In folk medicine, the genus *Eucomis*, including *Eucomis autumnalis* subspecies *autumnalis* (family: Asparagaceae), is a popular remedy against a variety of diseases [18]. For instance, it is administered as an enema to treat lower backache, biliousness, and urinary diseases, as well as fevers and fractures, among the Zulus in South Africa [19]. Recently, there have been increasing concerns about the conservation status of members in the genus *Eucomis* due to their endemic nature, indiscriminate harvesting, and wide utilization, particularly the underground parts such as the bulbs and roots [18]. The potential of using micropropagation as a means to ensure the sustainability of members of the genus has been recognized [18,20]. However, the quality and quantity of the bioactive phytochemicals in micro-propagated clones needs to be guaranteed to gain acceptance by different stakeholders, such as the consumers and traders in local communities, as well as herbal-based industries (nutraceutical and pharmaceutical companies) which are interested in these plants [21,22].

In most cases, extensively sought-after plants are often collected from the wild, resulting in a decline in their natural populations [17]. Researchers have recognized the need for effective conservation techniques for medicinal plants and devising approaches for the resupply of pharmacologically-active phytochemicals to meet the envisaged demands of the pharmaceutical industry [20,21,22]. Micropropagation generally allows for the mass production of clonal plant materials in a relatively short time and the utilization of elicitors to facilitate the accumulation of different phytochemicals [22]. In addition, an in vitro approach is often utilized to increase the biosynthesis and accumulation of antioxidant compounds in micropropagated plants [23]. Given that the value of cultivated medicinal plants is often dependent upon the quantity and quality of the accumulated phytochemicals which determine their bioactivity [21], research endeavors that can establish the integrity of micropropagated plants are desired. Thus, the current study evaluated the phytochemical content and antioxidant activity in in vitro regenerants and acclimatized *Eucomis autumnalis* subspecies *autumnalis* following treatment with SW and KAR_1_. The current study was guided by the following research questions:

(1) How does the application of SW and KAR_1_ influence the phytochemical pool and antioxidant activity of *Eucomis autumnalis* subspecies *autumnalis*?

(2) What are the dynamics of the aforementioned parameters in in vitro and acclimatized plants?

(3) What are the differences in terms of the phytochemical and antioxidant activities of the aboveground and underground parts of acclimatized plants?

## 2. Materials and Methods

### 2.1. Source of Chemicals and Plant Materials

Solutions of SW and KAR_1_ were obtained from the laboratory stock; details of their preparations have been extensively described [5,24]. We purchased the two internal standards (deuterium-labelled 4-hydroxybenzoic (2,3,5,6-D4) acid and salicylic (3,4,5,6-D4) acid) from Cambridge Isotope Laboratories (Andover, MA, USA). Solvents such as formic acid and methanol were supplied by Merck (Darmstadt, Germany). All chemicals used in the current study were of analytical standard.

*Eucomis autumnalis* subspecies *autumnalis* clones maintained on PGR-free media served as source of leaf explants for the current study. Mother stock of *Eucomis autumnalis* subspecies *autumnalis* was obtained from University of KwaZulu-Natal (UKZN, Pietermaritzburg, South Africa) Botanical Garden, and was positively identified (Voucher no: Masondo 2) by the curator of the Bews Herbarium, UKZN, South Africa.

### 2.2. In Vitro Propagation and Acclimatization Design

Leaf explants excised from stock plants were surface sterilized and cultured in tissue-culture screw-cap jars with 30 mL Murashige and Skoog (MS) media [25]. In order to generate the six (6) treatments, MS media was supplemented with SW at varying dilutions (1:500, 1:1000 and 1:1500) and KAR_1_ concentrations (10^−7^, 10^−8^ and 10^−9^ M). A control lacking SW and KAR_1_ was included in the experiment. Each treatment had 15 explants and the experiments was conducted in duplicate. The conditions of the growth room were similar to the previous setup as described by Masondo et al. [26]. After 10 weeks, in vitro regenerants were transferred for acclimatization in a mist house, and finally transferred again to a greenhouse for a duration of 4 months under the previously described conditions [11].

### 2.3. Ultra-High Performance Liquid Chromatography: Tandem Mass Spectrometry (UHPLC-MS/MS) Analysis of Phytochemicals

Plant material was harvested from in vitro regenerants and acclimatized *Eucomis autumnalis* subspecies *autumnalis* after 10 weeks and 4 months, respectively. In vitro (whole plantlets) and acclimatized plants separated into aerial (leaves) or underground (bulbs and roots) parts were oven-dried at 50 ± 2 ℃ for seven days and milled into powder form.

In triplicates, 20 mg of the ground plant samples were homogenized with 80% methanol (MeOH) using an oscillation ball mill (MM 301, Retsch, Haan, Germany) at a frequency of 27 Hz for 3 min. Deuterium-labelled internal standards (4-hydroxybenzoic (2,3,5,6-D4) acid and salicylic (3,4,5,6-D4) acid) were added to the extraction solvent prior to plant material homogenization. The resultant supernatant was centrifuged at 48,297.6 g for 10 min and retained for the quantification of the phytochemicals. Supernatants were filtered through 0.45 µm nylon membrane filters (Alltech, Breda, Netherlands) and analyzed using a UHPLC–MS/MS (Waters, Milford, MA, USA) linked to a Micromass Quattro microTM API benchtop triple quadrupole mass spectrometer (Waters MS Technologies, Manchester, UK), operating in a negative ion mode, as described by Gruz et al. [27] with modifications [28]. Authentic standards were used to identify and quantify all the phytochemicals. Due to the absence of authentic standards, eucomic acid (240.21 g/mol) was quantified as relative concentration (%) of control. The identification of eucomic acid was tentative, as it was based on mass spectra, UV profile, and the literature, as previously highlighted [28].

### 2.4. Plant Extraction and Antioxidant Activity Evaluation

Ground samples from in vitro and acclimatized *Eucomis autumnalis* subspecies *autumnalis* were extracted in 50% MeOH using 100 mg per 10 mL in an ultrasonic sonicator (Julabo GmbH, West Germany). This was sonicated for 20 min while maintaining the temperature with the use of ice-cold water. Following filtration, the resultant filtrates were dried to constant weight under a fan. The dried extracts were re-suspended in 50% MeOH and evaluated at a final concentration of 0.5 mg/mL in two test systems, namely, 2,2-diphenyl-1-picrylhydrazyl (DPPH) and *β*-carotene acid model, as detailed previously [29]. Ascorbic acid and butylated hydroxytoluene were used as positive controls in DPPH and *β*-carotene assays, respectively. In addition, 50% MeOH was included as the solvent control. Each experiment was done in triplicate.

### 2.5. Data Analysis

Experiments were conducted in completely randomized designs. Phytochemical content and antioxidant activity data were subjected to analysis of variance (ANOVA) using the SPSS software package for Windows (SPSS^®^, version 16.0, Chicago, IL, USA). Where there was statistical significance (*p* ≤ 0.05), the mean values were further separated using the Duncan’s Multiple Range Test (DMRT).

## 3. Results

### 3.1. Phytochemical Profiles of In Vitro Regenerants and Greenhouse-Acclimatized Plants

In total, 16 types of phytochemicals were accumulated in *Eucomis autumnalis* subspecies *autumnalis*. These consisted of eucomic acid, derivatives of hydroxybenzoic and hydroxycinnamic acids, as well as flavonoids.

#### 3.1.1. Eucomic Acid in In Vitro and Greenhouse-Acclimatized Plants

Both the in vitro and greenhouse-acclimatized *Eucomis autumnalis* subspecies *autumnalis* accumulated varying levels of eucomic acid, which were generally higher in the acclimatized plants compared to the in vitro regenerants (Figure 1). The major portion of the eucomic acid in the acclimatized plants was located in the aboveground organ (leaves) (Figure 1B). Relative to the controls, the application of SW or KAR_1_ had no significant effect on the level of the eucomic acid in the in vitro regenerants (Figure 1A) or the underground parts of the acclimatized plants (Figure 1C). However, SW (all three dilutions) and KAR_1_ (10^−8^ and 10^−9^ M) significantly enhanced the eucomic acid accumulated in the leaves of greenhouse-acclimatized plants (Figure 1B). The highest level of eucomic acid was accumulated in the leaves of plants treated with SW (1:1000), i.e., almost three times that in the control plants.

#### 3.1.2. Hydroxybenzoic Acid Derivatives In Vitro and Greenhouse-Acclimatized Plants

Four derivatives of hydroxybenzoic acids were accumulated in the different plant parts of *Eucomis autumnalis* subspecies *autumnalis* (Table 1). Only protocatechuic and *p*-hydroxybenzoic acids were present in in vitro (plantlets) regenerants and acclimatized (leaves, bulbs and roots) plants. The levels of these two derivatives were several times higher in in vitro plantlets than in the acclimatized plants. However, derivatives such as vanillic and syringic acids were specific to the in vitro regenerants and acclimatized (leaves) plants, respectively.

Among the in vitro regenerants, SW (1:500 and 1:1500 dilutions) and KAR_1_ (10^−9^ M) significantly enhanced the concentrations of protocatechuic and *p*-hydroxybenzoic acids. Likewise, in the acclimatized plants, both derivatives were significantly higher in KAR_1_ treatment for the leaves (10^−7^ M), bulbs, and roots (10^−8^ and 10^−9^ M) when compared to the control.

#### 3.1.3. Hydroxycinnamic Acid Derivatives in Vitro and Greenhouse-Acclimatized Plants

A total of five derivatives of hydroxycinnamic acids were quantified in the evaluated extracts of *Eucomis autumnalis* subspecies *autumnalis* (Table 2). However, only three derivatives, namely caffeic, coumaric, and ferulic acids, were present in the different plant parts after the acclimatization period. The quantity of hydroxycinnamic acid derivatives were relatively higher in the in vitro regenerants when compared to the acclimatized plants, irrespective of the plant parts and treatment regimes.

When compared to the control, treatments with either SW or KAR_1_ increased the concentrations of caffeic and cinnamic acids in in vitro regenerants. However, the concentrations of the other three derivatives were generally higher in the control than SW or KAR_1_ treatments. Isoferulic acid was present in the control, SW (1:1500) and KAR_1_ (10^−9^ M) treatments of the in vitro regenerants.

After acclimatization, the concentrations of caffeic and coumaric acids were generally low (<1 μg/g DW), regardless of the treatments. In addition, ferulic acid concentrations ranged from 0.17–0.39 and 0.78–1.89 μg/g DW for the leaves and underground parts, respectively. The higher concentrations of ferulic acid observed in the underground parts were facilitated by the presence of SW (1:1000 and 1:1500, dilutions) and KAR_1_ (10^−7^, 10^−8^ and 10^−9^ M).

#### 3.1.4. Flavonoids in Vitro and Greenhouse-Acclimatized Plants

Even though six types of flavonoids were quantified in *Eucomis autumnalis* subspecies *autumnalis*, their concentrations were generally low (0.1–3.4 μg/g DW), and only observed in a few treatments for both the in vitro and acclimatized plants (Table 3). Both hesperetin and kaempferol observed during the in vitro stage were not detected in any of the treatments or control after acclimatization. While eriodictyol was quantified in both underground and aboveground parts of the plant, other flavonoids were present in underground (pinobaksin) and aboveground (genistein and taxifolin) regions. Neither of the SW and KAR_1_ treatments had any stimulatory effect on the concentrations of flavonoids in the acclimatized plants.

### 3.2. Antioxidant Activity of In Vitro Regenerants and Greenhouse-Acclimatized Plants

The presence of SW and KAR_1_ significantly influenced the antioxidant potential of *Eucomis autumnalis* subspecies *autumnalis* in both DPPH (Figure 2) and *β*-carotene (Figure 3) assays. The free radical (DPPH) scavenging activity of the extracts from in vitro regenerants ranged from 26–55% and 23–74% for the acclimatized plants. For the DPPH test, SW (1:1000, dilution) and KAR_1_ (10^−9^ M)-treated in vitro regenerants had the highest antioxidant activity, i.e., approximately two-fold that of the control (Figure 2A). In addition, the majority (83%) of SW and KAR_1_-treated in vitro regenerants had significantly higher antioxidant activity compared to the control. However, only 50% (leaves) and 33% (roots and bulbs) of the ex vitro plants maintained a higher antioxidant activity than the control plants (Figure 2B,C). The leaves also generally had higher antioxidant activity than the underground plant parts among the acclimatized plants. 

In the *β*-carotene assay, antioxidant activity ranged from approximately 35–98% and 48–75% for the in vitro and ex vitro plants, respectively (Figure 3). The application of SW and KAR_1_ had minimum stimulatory effects on the antioxidant activity of in vitro regenerants and greenhouse-acclimatized *Eucomis autumnalis* subspecies *autumnalis* for the *β*-carotene test system (Figure 3). For instance, only 33% of the in vitro regenerants treated with KAR_1_ (10^−7^ and 10^−9^ M) had higher antioxidant activity than the control. About 67% of the leaves of ex vitro plants treated with SW or KAR_1_ had significantly higher antioxidant activity than the control, while there was no difference in the antioxidant activity of the roots and bulbs with or without the respective treatments (Figure 3B,C).

## 4. Discussion

Secondary metabolites in natural resources, including medicinal plants, have been widely explored for their biological properties [22,30], and members of the genus *Eucomis* are well-known for their rich phytochemicals [31]. Globally, there is concern for the increasing decline in valuable plants that are frequently collected from the wild, thereby causing severe strain and a decline of their natural populations [17,18,20]. The need for sustainability of plants as sources of valuable, bioactive compounds cannot be overemphasized [17,22]. Even though there is still limited knowledge on the plants biosynthetic pathway and underlying mechanisms of the action involved in the production of the desired phytochemicals [32], the use of elicitor(s) often influences their resultant integrity in terms of quality and quantity [33]. The potential role of SW and KAR_1_ on the phytochemical pool have been demonstrated in different plants, including *Musa* species [13], *Tulbaghia* species [34], *Isatis indigotica* [35], and *Aloe arborescens* [36]. In the current study, the inclusion of SW or KAR_1_ in the growth media during the micropropagation stage had a significant effect on the resultant phytochemicals in in vitro and acclimatized *Eucomis autumnalis* subspecies *autumnalis*. The therapeutic effects of the majority of the phytochemicals quantified in *Eucomis autumnalis* subspecies *autumnalis* are well established [22,23,37]. For example, ferulic acid is known to exhibit biological activities such as antioxidant and anti-inflammatory [38], and this particular compound was one of the major phenolic acids in the roots and bulbs of the acclimatized plants. The increased concentration of ferulic acid observed in the roots and bulbs of the acclimatized plants, obtained from SW (1:1000 and 1:1500, dilutions) and KAR_1_ (10^−7^, 10^−8^ and 10^−9^ M) treatments, is noteworthy, given that this plant is used for inflammation-related conditions.

The presence of an elicitor is known to activate genes related to defense-systems which often trigger the biosynthesis and accumulation of secondary metabolites [33]. This is also supported by the fact that in vitro propagation systems create some degree of abiotic stress. However, the transfer of the micro-propagated regenerants to ex vitro conditions is known to cause changes in the quality and quantity of secondary metabolites in plants [11,39,40]. This may be due to the utilization of some of the early-produced secondary metabolites as precursors for the biosynthesis and accumulation of other metabolites as the plant goes through different physiological stages over time [12,30,39]. The presence of higher concentrations of phenolic acids in in vitro regenerants than in acclimatized plants have been demonstrated in *Merwilla plumbea* [40], *Eucomis autumnalis* subspecies *autumnalis* [11], and *Artemisia judaica* [41]. Among the nine phenolic acids quantified in *Eucomis autumnalis* subspecies *autumnalis*, the concentrations of approximately 70% of the hyroxybenzoic (protocatechuic, *p*-hydroxybenzoic and vallinic acids) and hydroxycinnamic (coumaric, ferulic and cinnamic acids) derivatives were several times higher in the in vitro regenerants than in the acclimatized plants. However, eucomic acid was generally higher in the acclimatized plants, while no discrete pattern was observed with the concentrations of flavonoids at different plant stages.

Eucomic acid has been quantified in medicinal plants such as *Eucomis autumnalis* [28,42], *Opuntia ficus*-*indica* [43], and *Cryptostephanus vansonii* [44]. Previously, Okada et al. [45] isolated eucomic acid from *Lotus japonicus*; this was considered a potential leaf-opening factor (*LOF*) in this species. Likewise, the growth inhibitory potential of eucomic acid isolated from *Cattleya trianaei* was demonstrated at varying concentrations [46]. In the current study, eucomic acid was one of the phytochemicals that was abundant across the different treatments, regardless of the development stage. This observation suggests the wide distribution of eucomic acid in medicinal plants, and may be considered a marker compound in some species, especially for members of the genus *Eucomis*. Although no significant increase in the levels of eucomic acid was observed in in vitro plantlets treated with SW and KAR_1_, the leaves of the acclimatized plants, especially those from SW (1:500 and 1:1500) treatment, had significantly higher eucomic acid contents relative to the control. Despite the occurrence of eucomic acid in a large number of plants, especially in members of the genus *Eucomis*, evidence of their specific biological activities remains speculative.

Several pharmacological activities, including antioxidant potency, are often attributed to the quality and quantity of phenolic acids and flavonoids in plants [16,30,37]. Apart from being an important class of compound used for preventing many diseases, antioxidants play a crucial role as food additives to counteract spoilage caused by oxidizable nutrients [23,37]. The antioxidant activity of natural products is often evaluated via multiple methods that entail different mechanisms such as the hydrogen atom transfer (HAT) and single electron transfer (SET) [47,48]. As a result, the extracts of *Eucomis autumnalis* subspecies *autumnalis* were evaluated using DPPH and *β*-carotene assays in order to establish the influence of SW and KAR_1_ applications under diverse conditions. In the current study, the extracts generally demonstrated higher antioxidant potential in the *β*-carotene model than in the DPPH assay. The highest antioxidant power (*β*-carotene model) was observed in KAR_1_-treated in vitro regenerants, i.e., almost two-fold higher than the control. Based on the responses in the in vitro and acclimatized plants, whereby majority of the extracts had higher antioxidant activity in *β*-carotene model when compared to the DPPH assay, the mechanism of the antioxidant activity of *Eucomis autumnalis* subspecies *autumnalis* is more likely to be inclined towards HAT than SET. Similar findings were also exhibited by the extracts of micropropagated and acclimatized plants that were treated with different PGRs at varying concentrations [11,12,49]. Plants are generally known to synthesize and accumulate secondary metabolites at varying concentrations in their different organs [23], and this may influence the resultant antioxidant potential. However, it is often difficult to directly link the phytochemical pool to the antioxidant activity of medicinal plants [11,12,40,49].

In the current study, the variation observed in the antioxidant activity of different parts of acclimatized plants is important from a conservation perspective. For instance, the use of alternative plant organs, with a minimal detrimental effect on the survival of the whole plant, has been strongly recommended by researchers [17]. This means that the higher antioxidant activity exhibited by the leaves of acclimatized *Eucomis autumnalis* subspecies *autumnalis* provides a valuable alternative to the underground plant parts which are widely utilized in traditional medicine. The current findings also suggest that the biological effects of medicinal plants often differ based on the plant part investigated [10,18,44].

## 5. Conclusions

The importance of SW and KAR_1_ as potential elicitors for bioactive phytochemicals was demonstrated in *Eucomis autumnalis* subspecies *autumnalis*. This may provide an alternative approach for the production of secondary metabolites with therapeutic potential. Based on the use of UHPLC, the phytochemical profiles of *Eucomis autumnalis* subspecies *autumnalis* treated with SW and KAR_1_ was established for in vitro and acclimatized plants. Generally, in vitro regenerants accumulated higher concentrations of phytochemicals which significantly decreased after plants underwent prolonged periods of continuously-changing climatic conditions in the greenhouse. Among the nine phenolic acids in the in vitro regenerants, coumaric acid was the major (23-52 μg/g DW) bioactive compound. Acclimatized plants had only six types of phenolic acids, including syringic acid, which was absent in the in vitro stage. Likewise, the levels of a number of flavonoids were generally low, and different types were accumulated in in vitro and acclimatized plants. The levels of eucomic acid, which can be considered a diagnostic compound in *Eucomis* species, was significantly accumulated in the leaves of SW (1:1500) treatment after acclimatization. Antioxidant activity was relatively higher in the acclimatized plants when compared to the in vitro regenerants. Given the limitations associated with the two test systems used in the current study, the antioxidant activity demonstrated by the extract may be considered to be of low clinical significance. Thus, other test systems, especially in vivo systems, will be essential to reach a valid conclusion about the antioxidant potential of SW and KAR_1_-treated plants. From a conservation perspective, the current findings provide preliminary evidence of the value of SW and related technology as a potentially-viable method for the biosynthesis of phytochemicals of therapeutic importance in medicinal plants. Nevertheless, it will be necessary to establish the carry-over effect of SW and KAR_1_ for a longer duration (>1 year) on the phytochemical pools and other pharmacological activities (besides antioxidant), as well as to determine the overall safety of plant extracts. In addition, it will be necessary to investigate other medicinal plants in order to reach a valid conclusion about the overall potential of the tested compounds.

## Figures and Tables

**Figure 1 antioxidants-08-00611-f001:**
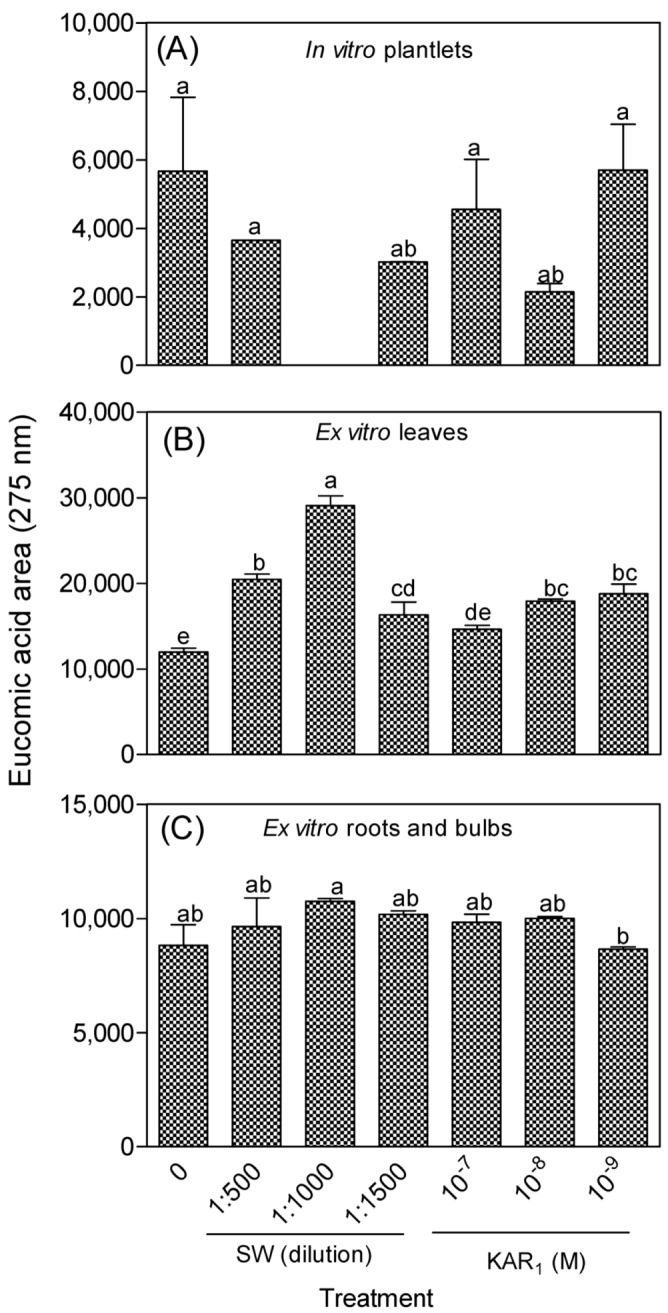
Accumulation of eucomic acid (area) in smoke-water (SW) and karrikinolide (KAR_1_)-treated *Eucomis autumnalis* subspecies *autumnalis*. (**A**): In vitro plantlet; (**B**) leaves and (**C**): roots and bulbs of greenhouse acclimatized plants. In each graph, bars with different letter(s) are significantly different (*p* ≤ 0.05) based on Duncan’s Multiple Test Range (DMRT), *n* = 3.

**Figure 2 antioxidants-08-00611-f002:**
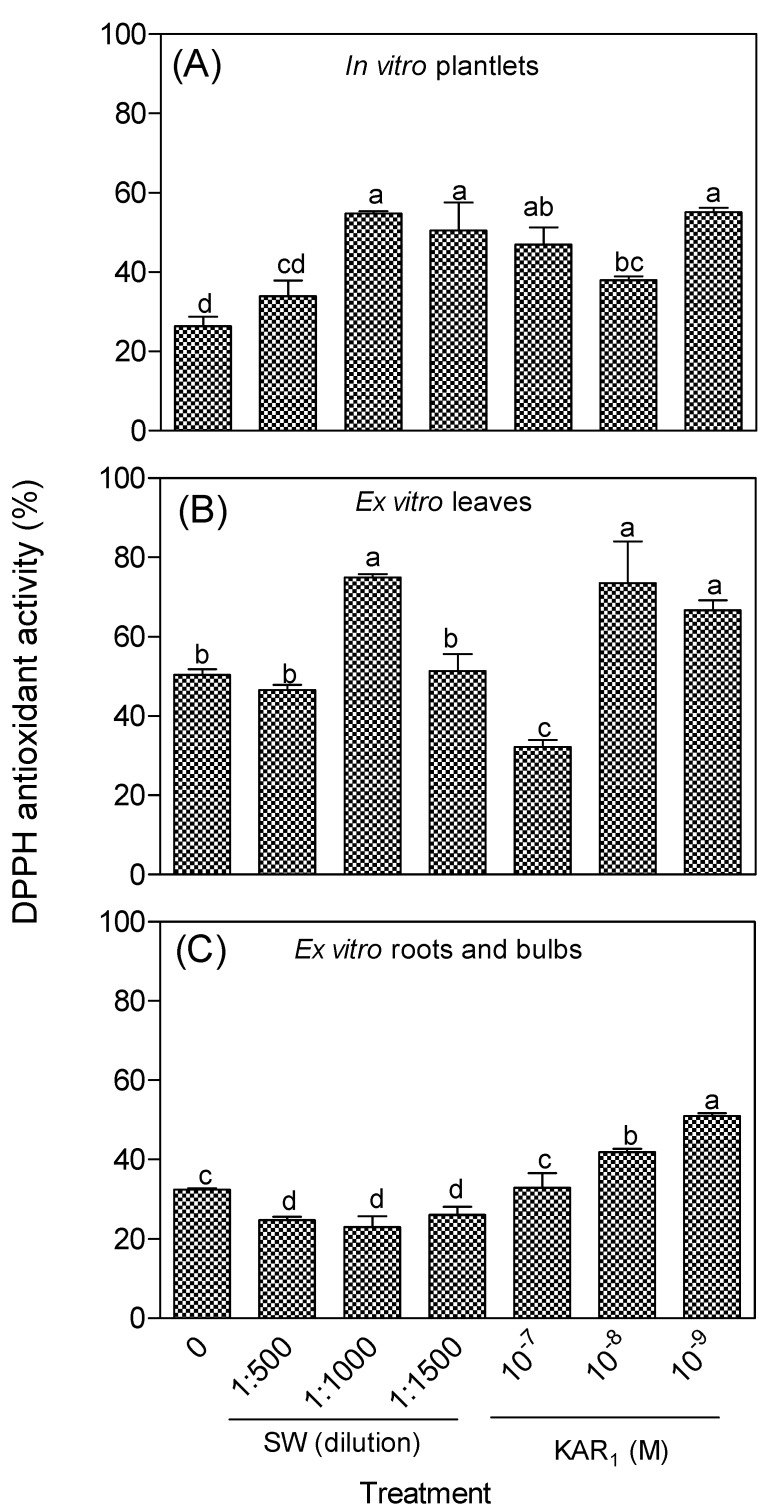
Effect of smoke-water (SW) and karrikinolide (KAR_1_) treatment on the free radical scavenging activity (2,2-diphenyl-1-picryhydrazyl, DPPH) of *Eucomis autumnalis* subspecies *autumnalis* (50% methanol) extract. (**A**): In vitro plantlet; (**B**): leaves; and (**C**): roots and bulbs of greenhouse acclimatized plants. In each graph, bar with different letter(s) are significantly different (*p* ≤ 0.05) based on Duncan’s Multiple Test Range (DMRT), *n* = 3. Value (%) for ascorbic acid (positive control) = 97.5 ± 0.03. All extracts and the positive control were evaluated at a final concentration of 0.5 mg/mL.

**Figure 3 antioxidants-08-00611-f003:**
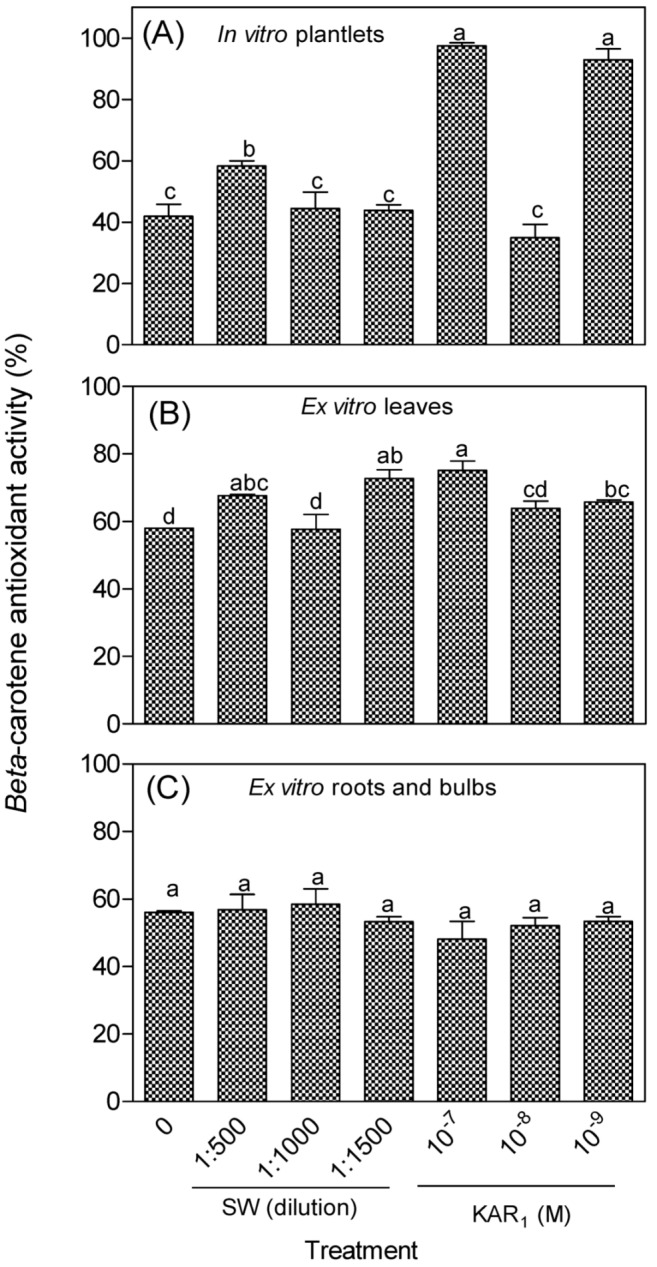
Effect of smoke-water (SW) and karrikinolide (KAR_1_) treatment on the antioxidant activity (*β*-carotene-linoleic acid model) of *Eucomis autumnalis* subspecies *autumnalis* (50% methanol) extract. (**A**): In vitro plantlet; (**B**): leaves; and (**C**): roots and bulbs of greenhouse acclimatized plants. In each graph, bar with different letter(s) are significantly different (*p* ≤ 0.05) based on Duncan’s Multiple Test Range (DMRT), *n* = 3. Value (%) for butylated hydroxytoluene, BHT (positive control) = 98.5 ± 0.06. All extracts and the positive control were evaluated at a final concentration of 0.5 mg/mL.

**Table 1 antioxidants-08-00611-t001:** Effect of smoke-water (SW, dilution) and karrikinolide (KAR_1_, M) on the concentrations (μg/g DW) of four hydroxybenzoic acid derivatives in different plant parts of in vitro and greenhouse-acclimatized *Eucomis autumnalis* subspecies *autumnalis*.

Plant Stage/Part	Treatment	Protocatechuic Acid	*p*-Hydroxybenzoic Acid	Vanillic Acid	Syringic Acid
In vitro	Control	1.7 ± 0.02 d	14.5 ± 0.43e	1.7 ± 0.27a	nd
Plantlets	SW 1:500	2.1 ± 0.01 c	21.2 ± 0.19b	1.8 ± 0.09a	nd
	SW 1:1000	nd	nd	nd	nd
	SW 1:1500	2.6 ± 0.03a	28.7 ± 0.22a	1.7 ± 0.01a	nd
	KAR_1_ 10^−7^	1.6 ± 0.04e	18.7 ± 0.41d	1.4 ± 0.18a	nd
	KAR_1_ 10^−8^	1.7 ± 0.01de	14.6 ± 0.28e	1.9 ± 0.10a	nd
	KAR_1_ 10^−9^	2.3 ± 0.04b	20.0 ± 0.30c	1.5 ± 0.20a	nd
Ex vitro	Control	0.6 ± 0.02c	0.5 ± 0.01b	nd	nd
Leaves	SW 1:500	1.0 ± 0.03b	0.3 ± 0.01c	nd	nd
	SW 1:1000	0.4 ± 0.00d	0.3 ± 0.01c	nd	nd
	SW 1:1500	0.6 ± 0.01c	0.5 ± 0.02b	nd	nd
	KAR_1_ 10^−7^	1.1 ± 0.01a	0.8 ± 0.05a	nd	nd
	KAR_1_ 10^−8^	0.2 ± 0.00e	0.3 ± 0.01c	nd	nd
	KAR_1_ 10^−9^	0.7 ± 0.05c	0.6 ± 0.12b	nd	nd
Bulbs + roots	Control	0.8 ± 0.09c	0.8 ± 0.01d	nd	0.2 ± 0.03b
	SW 1:500	0.3 ± 0.03f	0.8 ± 0.02e	nd	0.1 ± 0.02c
	SW 1:1000	0.6 ± 0.02d	0.9 ± 0.05c	nd	0.1 ± 0.03b
	SW 1:1500	0.5 ± 0.02de	0.7 ± 0.01d	nd	0.1 ± 0.01abc
	KAR_1_ 10^−7^	0.5 ± 0.00e	0.7 ± 0.00e	nd	0.2 ± 0.01ab
	KAR_1_ 10^−8^	1.0 ± 0.04b	1.0 ± 0.02b	nd	0.1 ± 0.01abc
	KAR_1_ 10^−9^	1.1 ± 0.02a	2.7 ± 0.03a	nd	0.2 ± 0.04a

In each column, plant part with different letter(s) are significantly different (*p* ≤ 0.05) based on Duncan’s Multiple Test Range (DMRT), *n* = 3. nd = not detected.

**Table 2 antioxidants-08-00611-t002:** Effect of smoke-water (SW, dilution) and karrikinolide (KAR_1_, M) on the concentrations (μg/g DW) of five hydroxycinnamic acid derivatives in different plant parts of in vitro and greenhouse-acclimatized *Eucomis autumnalis* subspecies *autumnalis*.

Plant Stage/Part	Treatment	Caffeic Acid	Coumaric Acid	Cinnamic Acid	Ferulic Acid	Isoferulic Acid
In vitro	Control	0.20 ± 0.030d	51.55 ± 0.562a	3.3 ± 0.24d	11.82 ± 0.246a	0.9 ± 0.18a
plantlet	SW 1:500	0.36 ± 0.007c	29.78 ± 0.295c	13.5 ± 0.05a	3.39 ± 0.019d	nd
	SW 1:1000	nd	nd	nd	nd	nd
	SW 1:1500	0.56 ± 0.008a	39.19 ± 1.163b	10.9 ± 0.06 b	6.73 ± 0.020b	0.9 ± 0.05a
	KAR_1_ 10^−7^	0.46 ± 0.045b	31.91 ± 0.215c	11.5 ± 0.82 b	4.56 ± 0.082c	nd
	KAR_1_ 10^−8^	0.56 ± 0.013a	23.23 ± 0.528d	7.8 ± 0.75 c	3.54 ± 0.150d	nd
	KAR_1_ 10^−9^	0.34 ± 0.044c	38.92 ± 1.665b	10.3 ± 0.29 b	4.51 ± 0.242c	0.5 ± 0.01b
Ex vitro	Control	0.09 ± 0.005d	0.76 ± 0.102bc	nd	0.28 ± 0.010bc	nd
leaves	SW 1:500	0.16 ± 0.007c	0.66 ± 0.054cd	nd	0.36 ± 0.002ab	nd
	SW 1:1000	0.26 ± 0.007a	0.64 ± 0.049cd	nd	0.17 ± 0.022d	nd
	SW 1:1500	0.13 ± 0.010c	0.89 ± 0.068b	nd	0.24 ± 0.049cd	nd
	KAR_1_ 10^−7^	0.13 ± 0.016c	1.19 ± 0.071a	nd	0.34 ± 0.007ab	nd
	KAR_1_ 10^−8^	0.16 ± 0.012c	0.50 ± 0.066de	nd	0.39 ± 0.048a	nd
	KAR_1_ 10^−9^	0.21 ± 0.013b	0.40 ± 0.015e	nd	0.18 ± 0.018d	nd
Bulbs + roots	Control	0.14 ± 0.004a	0.31 ± 0.007cd	nd	0.78 ± 0.046f	nd
	SW 1:500	0.07 ± 0.009c	0.25 ± 0.019e	nd	0.82 ± 0.010ef	nd
	SW 1:1000	0.12 ± 0.010ab	0.32 ± 0.008c	nd	1.37 ± 0.060b	nd
	SW 1:1500	0.13 ± 0.002ab	0.28 ± 0.003de	nd	0.96 ± 0.035de	nd
	KAR_1_ 10^−7^	0.12 ± 0.012ab	0.27 ± 0.006de	nd	1.02 ± 0.021cd	nd
	KAR_1_ 10^−8^	0.11 ± 0.006b	0.42 ± 0.023b	nd	1.14 ± 0.063c	nd
	KAR_1_ 10^−9^	0.12 ± 0.001ab	0.50 ± 0.008a	nd	1.89 ± 0.067a	nd

In each column, plant part with different letter(s) are significantly different (*p* ≤ 0.05) based on Duncan’s Multiple Test Range (DMRT), *n* = 3. nd = not detected.

**Table 3 antioxidants-08-00611-t003:** Effect of smoke-water (SW, dilution) and karrikinolide (KAR_1_, M) on the concentrations (μg/g DW) of six flavonoids in different plant parts of in vitro and greenhouse-acclimatized *Eucomis autumnalis* subspecies *autumnalis*.

Plant Stage/Part	Treatment	Hesperetin	Kaempferol	Eriodictyol	Genistein	Pinobaksin	Taxifolin
In vitro	Control	nd	nd	nd	nd	nd	nd
Plantlet	SW 1:500	nd	nd	nd	nd	nd	nd
	SW 1:1000	nd	nd	nd	nd	nd	nd
	SW 1:1500	nd	1.7 ± 0.09	nd	nd	nd	nd
	KAR_1_ 10^−7^	nd	nd	nd	nd	nd	nd
	KAR_1_ 10^−8^	nd	nd	nd	nd	nd	nd
	KAR_1_ 10^−9^	3.4 ± 1.22	nd	nd	nd	nd	nd
Ex vitro	Control	nd	nd	0.18 ± 0.008a	0.05 ± 0.008a	nd	nd
Leaves	SW 1:500	nd	nd	0.11 ± 0.004b	nd	nd	nd
	SW 1:1000	nd	nd	0.07 ± 0.001c	0.03 ± 0.004b	nd	nd
	SW 1:1500	nd	nd	nd	nd	nd	0.17 ± 0.039
	KAR_1_ 10^−7^	nd	nd	nd	nd	nd	nd
	KAR_1_ 10^−8^	nd	nd	nd	nd	nd	nd
	KAR_1_ 10^−9^	nd	nd	nd	nd	nd	nd
Bulbs + roots	Control	nd	nd	0.23 ± 0.015a	nd	0.12 ± 0.020a	nd
	SW 1:500	nd	nd	0.13 ± 0.017b	nd	0.07 ± 0.004b	nd
	SW 1:1000	nd	nd	0.10 ± 0.019b	nd	nd	nd
	SW 1:1500	nd	nd	nd	nd	nd	nd
	KAR_1_ 10^−7^	nd	nd	nd	nd	nd	nd
	KAR_1_ 10^−8^	nd	nd	nd	nd	nd	nd
	KAR_1_ 10^−9^	nd	nd	nd	nd	nd	nd

In each column, plant parts with different letter(s) are significantly different (*p* ≤ 0.05) based on Duncan’s Multiple Test Range (DMRT), *n* = 3. nd = not detected.

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
