# Peer review of "Potential of Smoke-Water and One of Its Active Compounds (karrikinolide, KAR1) on the Phytochemical and Antioxidant Activity of Eucomis autumnalis"

_antioxidants, 2019, doi:10.3390/antiox8120611_

Round 1

Reviewer 1 Report

The manuscript id focused on the determination of the effect of smoke-water and karrikinolide on some chemical constituents and antioxidant activity of in vitro and greenhouse-acclimatized Eucomis autumnalis subsp. autumnalis.
Phytochemicals (eucomic acid, 15 phenolic acids, and flavonoids) were determined in different levels depending on the source of plant material and experimental treatments. Considering the importance of E. autumnalis as a medicinal plant and a risk of this species extinction, the biochemical evaluation of in vitro regenerants and acclimatized plants is highly desirable. Moreover, smoke water and karrikinolide are newly recognized elicitors of medicinal plants' chemical constituents. In my opinion, the manuscript has undisputable merits and represents novelty. It’s written according to high standards of scientific elaborations. I recommend to accept it for publication after minor revision concerning technical aspects.

Reviewer 2 Report

The submitted paper describes an interesting experiment dealing with the conditions of plant reproduction/cultivation and the concentration of selected antioxidant compounds.

Compounds identification is correctly carried out, by using different and independent parameters.

Antioxidant activity was evaluated trough two well established methods.

I just suggest to precise the temperature and time used for samples drying that can be a limiting factor of the concentration of antioxidants.

Reviewer 3 Report

The authors, in the present study, have quantitatively and qualitatively evaluated phytochemical and antioxidant activity of  Eucomis  autumnalis following treatment with smoke-water and karrikinolide . The work is technical sound and the authors utilized appropriate techniques of analysis, but, in my opinion, the quality of the work is very low and the data are two preliminary and not so relevant as the authors state. The analysis of antioxidant activity is very poor and need clearly to be amplified performing other tests (such as FRAP, TEAC and ORAC, just to mention a few), because the two performed tests just supply an idea, and, as authors known, they follow a specific mechanism of elimination, so it is inappropriate, today, to speak of antioxidant assay based only on these two assays.  Moreover the increase in antioxidant potential not specifically is linked with health promoting properties, because the analyzed substances have either an antioxidant and a pro-oxidant activity inside the organisms, so a test to evaluate their potential on cell culture or in vivo, may be useful before starting to draw some conclusions. In the conclusion, in particular, authors have to pay particular attention in drawing so general conclusion “From a conservation perspective, the current findings  provide evidence on the value of SW and related technology as a viable method for biosynthesis of phytochemicals of therapeutic importance in medicinal plants”, because they have evaluated just Eucomis  autumnalis, and what is true for this plants it is not sure for other species. It is more appropriate to take into consideration their results.

Round 2

Reviewer 3 Report

Dear authors I don’t know how you can do to obtain other samples, but in my opinion, it is not enough to write just some sentences, because the two assays performed in the work are not enough to speak of hydrogen atom transfer (HAT) reaction and (2) single electron transfer (ET) reaction. The two assays you have performed are an example of the two mechanisms, but we have to utilize more assays because that the sample work with an essay that is based on, for instance,  hydrogen atom transfer (HAT) reaction do not imply that it works with other assays based on the same mechanism. This is of particular interest for your work that has as one of the main aim the analysis of antioxidant activity. So I’m afraid, but the work is not suitable in the present form.
